# Towards an Innovative Sensor in Smart Capsule for Aerial Drones for Blood and Blood Component Delivery

**DOI:** 10.3390/mi13101664

**Published:** 2022-10-03

**Authors:** Rongrong Liu, Giorgio Pitruzzello, Mafalda Rosa, Antonella Battisti, Chiara Cerri, Giuseppe Tortora

**Affiliations:** 1BioRobotics Institute, Scuola Superiore Sant’Anna, 56127 Pisa, Italy; 2Department of Excellence in Robotics & AI, Scuola Superiore Sant’Anna, 56127 Pisa, Italy; 3Smart Medical Theatre Laboratory, ABzero, 56124 Pisa, Italy; 4Istituto Nanoscienze—CNR and Scuola Normale Superiore, 56127 Pisa, Italy; 5Department of Pharmacy, University of Pisa, 56126 Pisa, Italy

**Keywords:** drone delivery, blood components transport, thermal stability, haemolysis detection

## Abstract

Aerial drone technology is currently being investigated worldwide for the delivery of blood components. Although it has been demonstrated to be safe, the delivered medical substances still need to be analyzed at the end of the flight mission to assess the level of haemolysis and pH prior to the use in a patient. This process can last up to 30 min and prevent the time saved using drone delivery. Our study aims to integrating an innovative sensor for the haemolysis and pH detection into the Smart Capsule, an already demonstrated technology capable of managing transfusion transport through drones. In the proposed scenario, the haemolysis is evaluated optically by a minilysis device using LED–photodetector combination. The preliminary validation has been demonstrated for both the thermal stability of the Smart Capsule and the haemolysis detection of the minilysis device prototype. Firstly, the onboard temperature test has shown that the delivery system is capable of maintaining proper temperature, even though the samples have been manipulated to reach a higher temperature before inserting into the Smart Capsule. Then, in the laboratory haemolysis test, the trend of linear regression between the outputs from the spectrophotometer and the minilysis prototype confirmed the concept design of the minilysis device.

## 1. Introduction

Blood and blood components represent lifesaving public goods, which are included in the World Health Organization (WHO) model list of essential medicines [1]. Blood delivers oxygen and nutrients to organs and tissues, removes waste, helps fight infections and heal from injuries, transports messages with the transport of hormones and the signaling of tissue damage, regulates core body temperature, etc. The average adult has about 5 L of blood circulating inside their body, and it is composed of blood cells suspended in blood plasma. The blood cells are mainly red blood cells (RBCs, or erythrocytes), white blood cells (WBCs, or leukocytes) and platelets (or thrombocytes) [2]. Specifically, RBCs contain a protein called hemoglobin, which allows RBCs to deliver oxygen from the lungs to the other tissues around the body. RBCs have a lifespan of approximately 120 days. WBCs are part of the body’s immune system. Different types of WBCs exist, such as neutrophils, lymphocytes, monocytes, eosinophils and basophils, which can be alive for hours, days, months, or years depending on the specific type. Platelets are cell fragments without a nucleus that help with clotting. Platelets live in the body for 7 to 10 days. Plasma is the liquid where the blood cells are immersed, and it constitutes 55% of the total blood volume, which is made up mostly of water, together with proteins, glucose, mineral ions, fatty substances, salt, nutrients, vitamins and hormones.

Blood and blood components are used daily in hospitals around the world. They are transported to the places and times when they are needed. However, the high perishability and scarcity of blood products make their availability and timeliness critical. Aerial drone technology has been investigated and exploited worldwide to improve medical care to save lives, particularly with the delivery of blood components. One of the most remarkable achievements is the blood-carrying drone operated by the USA company Zipline in the deep rural areas of Rwanda, where the terrain is characterised by impassable mountains and damaged roads [3]. Meanwhile, Swiss Post, together with Matternet, has focused on the transportation of blood samples between hospitals or laboratories located in urban regions, in which drones have achieved a very competitive delivery time especially when the ground deliveries are affected by congestion [4]. Moreover, other efforts have also been devoted in Japan [5], Canada [6], Italy [7,8] and China [9], etc.

One of the difficulties in using drones is to maintain blood components transported under critical thermal conditions, as they are easy to spoil if they are stored out of the temperature range or in improper conditions [8]. For instance, RBCs must be maintained between 2 °C and 6 °C [10], while the quality of plasma constituents is best maintained in a frozen state and platelet storage is optimal at room temperature with continuous agitation. This problem is partially overcome by the use of thermally insulating containers and by taking out the advantage of the short average duration of the flight in the aforementioned cases.

Furthermore, another problem in the case of whole blood and RBCs transportation is the lack of information concerning the quality of blood, such as haemolysis, which is the destruction of red blood cells and the release of hemoglobin into surrounding plasma. The fundamental consequence of haemolysis is a decrease in the ability to carry oxygen. Furthermore, after the hemolytic process, free plasma hemoglobin interacts with many metabolites and is associated with an increase in conjugated and unconjugated bilirubin, as well as an increase in lactate dehydrogenase. Hemolysis may occur not only in vivo but also in vitro [11]. Specifically, haemolysis inside the body can be caused by a large number of medical conditions [12], such as Gram-positive bacteria [13], parasites [14], autoimmune disorders [15], genetic disorders [16], or blood with an insufficient solute concentration [17], while in vitro haemolysis may result from the improper techniques during the collection of blood specimens [18], the effects of mechanical processing of blood [19], or bacterial contaminations [20]. Various photospectrometric approaches for clinics and portable devices have been investigated to measure hemoglobin concentration [21]. Measurement of hemoglobin concentration in blood is currently one of the most frequently performed clinical laboratory tests [22,23], by comparing the specimen color with a color chart after centrifugation. However, measurement of free hemoglobin content in plasma, which is derived from RBCs haemolysis and can be used to evaluate the vitro haemolysis in blood bags, is more difficult due to the low concentrations [24]. Thus, automatic instruments based on spectroscopic, laser or photometric techniques [25] are available in the market.

Our project relies on two main research lines, Namely, the development of an intelligent container equipped with a temperature control unit, which is capable of managing transport through drones, and the development of a low-cost implementable device that provides the quantitative monitoring of whole blood and RBCs during the process of both storage and transport. We aim to integrate this miniaturized minilysis device for the optical detection of haemolysis into the Smart Capsule, which is an autonomous device for the drone delivery of medical materials [7]. As a modular device, the Smart Capsule can integrate an additional sensor on board that could potentially increase the application of drone delivery worldwide, making the real time quality assessment on board possible for the first time ever in a compact device rather than bulky laboratory machines. The integration of a haemolysis and pH sensor will be of utmost importance for the future of drone delivery. Making an early quality assessment of haemolysis and pH during a drone delivery mission can help exploit this technology in future, without the need for additional tests of haemolysis and pH at the final destination before the use on a patient. The implementable minilysis device is activated to evaluate haemolysis during the transportation process thanks to its connection capabilities. In this sense, the condition of blood bags can be monitored and guaranteed in a secure manner and with a high reliability in real time, especially for a test lasting more than 30 min where it makes up for the difference in transportation time and raises questions about the viability of drone transport. Although drone delivery has been demonstrated to be safe for the delivery of medical materials [26], haemolysis tests should still be performed after drone landing. This results in a 30–45 min delay.

The scope of this study is to develop innovative haemolysis and pH sensor for the real time monitoring of blood components quality in drone delivery. In this manuscript, as a first step, experiments will be conducted to confirm the ability of the Smart Capsule in maintaining the temperature in the desired range during flight and validate the effectiveness of the minilysis device in a laboratory compared to current laboratory technologies. The structure of the manuscript is organized as follows. In Section 2, the materials and methods are explained in detail, including the Smart Capsule and minilysis, together with blood preparation for the haemolysis test in the laboratory. The results of both the onboard temperature test and the laboratory haemolysis test are presented in Section 3. In Section 4, the discussion of this study and some ideas for future work on the project are proposed.

## 2. Materials and Methods

### 2.1. Smart Capsule

The Smart Capsule is an innovative intelligent container of polyurethane specifically designed for the transportation of perishable and high-value blood as well as other medical products such as organs, tissues, swabs, test samples, drugs, etc., which need to be maintained under strict conditions and rapidly delivered. Figure 1 displays how the Smart Capsule is embedded on a drone with a dedicated adjustable interface to adapt to the continuous evolution of drones. This Smart Capsule is equipped with an artificial intelligence (AI) module able to take over the control of the drone it is mounted on, or to make it redundant, thereby guaranteeing double insurance for successful task completion. Additionally, an intuitive mobile app is also available to interact with the drone control system and monitor the delivery process. For more details about the hardware and software, please refer to our other previous publications [7,8].

Several efforts have been undertaken to meet the temperature-control requirements for the transport and storage at a constant temperature of heat-sensitive blood components. Firstly, the polyurethane envelope of the Smart Capsule contributes to maintaining proper temperatures to some extent. Secondly, a certified container UN3373 is equipped, following the delivery of blood and blood components by traditional means. Thirdly, temperature stabilizers are adopted, which are made of high- density polyethelene (HDPE) and contain a solution of water and paraffin mixtures. That is to say, these temperature stabilizers are cooled in advance to a suitable temperature before being inserted in the UN3373 container, together with the blood components. Furthermore, a temperature control unit has been implemented, as shown in Figure 2. Given a certain delivery mission, which comprises the precise thermal condition $T_{int}$, the time limit $t_{max}$ and the landing destination, the energy provided for the drone engine $E_{eng}$ can be predicted. With the two temperature sensors configured to measure the internal temperature $T_{int}$ and external temperature $T_{ext}$, the energy $E_{term}$ can be calculated to maintain the thermal unit at $T_{int}$ during the flight. This control unit monitors whether the residual energy $E_{res}$, including other auxiliary energy sources $E_{aux}$, is enough to support the mission. Otherwise, the delivery task needs to be modified. Although temperature is one of the basic parameters used to guarantee during drone delivery, the haemolysis level of the blood bag must be tested at the final destination, with the consequent waste of time and resources. For this reason, the minilysis device for innovative haemolysis and pH sensing has been proposed for the real time monitoring of blood components’ quality in drone delivery.

### 2.2. Minilysis

The minilysis has been proposed and designed to be a low-cost implementable device from scratch, which is an alternative to the existing expensive and cumbersome machines equipped in hospitals for the optical detection of haemolysis. Essentially, the working principle of the device relies on the quantification of the photons absorbed (or transmitted) by the liquid sample when exposed to light at a specific wavelength. After proper calibration, the device will be able to correlate the output voltage, which depends on the transmitted light, with hemoglobin concentration in the sample. In the prototype presented in Figure 3, a beam of light with a wavelength of 540 nm is sent from the light-emitting diode (LED) emitter on top, through the specimen held in a commercial Diaspect cuvette, and to the photodetector receiver on the bottom. The wavelength has been selected based on the fact that one of the strongest absorption peaks of hemoglobin falls in the 540–600 nm range [27,28,29]. By detecting the residual intensity of light after absorption, the percentage of hemoglobin inside the specimen can be evaluated. That is to say, the higher haemolysis with more hemoglobin in the specimen, the smaller the voltage read in the output circuit. After some calibration tests performed by comparing the minilysis output signals with those obtained spectrophotometrically from the same set of samples, quantification of released hemoglobin can be achieved. A more mechanically compact version is being designed in the lab with the same principle, as shown in Figure 4, with a dimension of 6 × 3 × 1.6 cm, where the upper part is implemented to support the LED emitter, while the bottom part is used for the photodetector receiver and the corresponding circuit.

### 2.3. Blood Preparation for Laboratory Haemolysis Test

The blood sample for the measurements was collected from donation in the Apuane Civil Hospital (NOA) in Massa in Italy, where a patient with excessive hematocrit consented to the use of his blood during the bloodletting therapy for research purposes, as shown in Figure 5. Informed consent has been acquired from an ethical point of view, all methods have been carried out in accordance with relevant guidelines and regulations, and all experimental protocols have been approved by the Ethical Committee of Azienda Sanitaria Locale Toscana Nord Ovest (ASLTNO) and the Heart Hospital Gaetano Pasquinucci in Massa, Italy.

All the bags used during blood preparation were first sterilized in an autoclave at 115 °C for 30 min, and then kept at temperatures not exceeding 40 °C while protected from frost and light. The donation bag was then sent to the Transfusion Center in the Pisa University Hospital Cisanello, where the blood was separated with a squeezing instrument following the traditional procedure to three bags containing plasma, a buffy coat (leukocyte and platelets), and concentrated RBCs, respectively, as shown in Figure 6.

After this separation procedure, only the bag containing concentrated RBCs was transported to the laboratory in the Heart Hospital Gaetano Pasquinucci in Massa. The blood has been divided into 7 sub-bags with a volume of 30 mL, which would provide as much data as possible with a series of tests, and make it possible to analyze the behavior of different sub-bags over time. All the blood bags were stored in the fridge at 4 ${}^{{}^{\circ}}C$.

## 3. Experiments and Results

### 3.1. Onboard Temperature Test

The first phase of this study was performed in collaboration with the ABzero spin off company of Scuola Sant’Anna and the municipality of Pontedera (PI), Italy, by performing a first preliminary flight. The system was proved to be intuitive and easy to use by the medical staff, as demonstrated during the tests in which the autonomous flight was activated by the hospital personnel. The core of the technology is the intelligent capsule equipped with temperature control and traceability. It proved effective in maintaining the correct range of temperatures and also versatile when used as a self-standing device to temporarily store medical goods before the flight.

The first experimental test was dedicated to assessing the thermal stability of the Smart Capsule for drone delivery. In this case, bags with non-biological materials, namely physiological solutions, were inserted into the Smart Capsule. These tests were repeated for routes of around 15 min for a total distance of 105 km and 39 h flight time in a territory with different geographical characteristics with a maximum height of 150 m and a maximum speed of 60 km/h. The temperature variation of non-biological materials during all flights was recorded and one of them was illustrated as a representative in Figure 7, from the time the samples were taken out from the fridge to the end of the flight. The samples were first manipulated until they reached 11 °C, which was a conventional operation implemented to create a more challenging experimental situation. Subsequently, they were inserted into the Smart Capsule around 30 min before the flight started. It is clear that the delivery system is an effective way to cool down the samples and to maintain the temperature in the desired range during flight.

### 3.2. Laboratory Haemolysis Test

After the arrival of the RBCs, haemolysis detection tests were conducted in Heart Hospital Gaetano Pasquinucci. To follow the haemolysis progression of the blood bag, we performed a series of 9 tests every 4 days on a small amount of blood taken from the same blood bag. According to the relevant regulation, blood donation bags have to be thrown away 42 days after collection. Thus, it was not necessary to exceed this period. The blood samples were properly discarded of after each experimental session.

The test procedures for the Ultrospec 2000 spectrophotometer are as follows: a blood sample of 1 mL was extracted from each sub-bag, put inside of test tubes labeled from 1 to 7, and centrifuged with a rate of 2000 g for 10 min by Eppendorf 5810R centrifuge to remove any residual RBCs; a 1 mL $Na_{2}CO_{3}$ solution stored in the lab was first pipetted into each cuvette labeled as blank and test 1 to 7, respectively; then, 0.1ml of $Na_{2}CO_{3}$ the solution was firstly pipetted to the blank cuvette, and 0.1 mL of the specimen plasma of 1 to 7 was then added to the corresponding cuvettes, obtaining an 11-fold dilution of the plasma sample; the Ultrospec 2000 spectrophotometer was activated and calibrated with the blank cuvette; the cuvettes with specimen plasma were also placed inside the spectrophotometer and the absorbance at wavelengths of 415, 450 and 700 nm was measured and recorded for 7 cuvettes in sequence; the total hemoglobin concentration in mg/dL was calculated manually with the following empiric Equation (Equation 1).
(1)$$Hb=154.7\times A_{415}-130.7\times A_{450}-123.9\times A_{700}$$
where $A_{415}$, $A_{450}$ and $A_{700}$ represent the absorbance at wavelength of 415, 450 and 700 nm, respectively [25]. Importantly, for any absorbance reading that exceeds 2.0, another 1:11 dilution needs to be reduced to within the accurate range of the spectrometer. Accordingly, the calculated result will be multiplied by the dilution factor of 11. During our test, this procedure was necessary starting from the 7th test for some sub-bags with a higher haemolysis level.

Figure 8 illustrates the haemolysis for each sub-bag in the sequence of nine tests. It is obvious that the haemolysis level increases with time for all sub-bags, especially for Sub-bags 2 and 7 between the last two tests, where the hemoglobin concentration increased considerably. It is also noted that the initial haemolysis levels are not the same for the seven sub-bags, even though they came from the same donor and experienced the same blood component separation process. Based on these results, the authors speculated that this difference was induced by the mechanical procedure dividing the whole bag to seven sub-bags.

When we observed the development of the hemoglobin concentration more carefully, we also noticed some abnormal values for sub-bags 2 and 6 on the seventh test. In fact, it was due to an operational error. As we stated earlier, when the reading of the spectrophotometer was higher than 2.0, a second 1:11 dilution was needed. From the seventh test, the absorbance readings at a wavelength of 415 nm for these two sub-bags started to exceed 2.0, specifically reaching up to 2.398 and 2.235. We kept these two datapoints here for the sake of truth and completeness, also as a reminder for future tests.

On the last test day, an extra dilution test using a physiological solution was designed and conducted with a specimen from sub-bag 4, 5 and 6. The dilution rates were chosen as 3:4, 2:3, 1:2, 1:3 and 1:4, considering the sufficiency and easy operation. The corresponding hemoglobin concentration of the diluted specimens was measured in the same manner described above and illustrated in Figure 9. The linearity in the range of 70 to 500 mg/dL for all three specimens shows a good stability and accuracy of this measure method, which confirms that it is sufficient to act as a quantitative reference for our ongoing development of a low-cost implementable device to evaluate haemolysis.

The measurement with the minilysis device was also conducted for all the original specimens and diluted ones on this last test day. Following the centrifugal process for the spectrophotometer test, the specimen was pipetted into the commercial Diaspect cuvette and inserted between the LED and photodiode. Once minilysis device was turned on, and the corresponding software for Arduino could read the output voltage and store automatically. With the detection results for the same specimen obtained from the spectrophotometer and minilysis device, a linear regression was carried out and is illustrated in Figure 10. Despite some fluctuations (with a standard deviation of 157.31 mV for the minilysis output), the data clearly show an inverse correlation between the output voltage read by the minilysis and the increasing hemoglobin concentration detected spectrophotometrically. Statistically, the correlation coefficient is −0.8085, with *p* < 0.0001, which shows the corresponding correlation between the outputs of minilysis and spectrophotometer is considered significant. As expected, the output voltage decreases as haemolysis goes on, due to increasing light absorption by progressively released hemoglobin. This trend confirms the design of minilysis as a suitable strategy to follow the hemolytic process over time. Concerning the fluctuations, the authors found that the mechanical support in this prototype version to fix the cuvette does not perfectly fit the shape of the cuvette; thus, it is difficult to maintain the same position each time a new cuvette is inserted. This aspect can be easily fixed and will be taken into account for the realization of the final device.

## 4. Discussion on Results

This study aims to integrate a miniaturized minilysis device for the optical detection of haemolysis into a Smart Capsule capable of managing transport through drones, which is the first step towards the development of a bio-Smart Capsule able to maintain the conditions of living cells implementing AI based on real living cells, such as neurons, cardiac and other cells. In this manuscript, preliminary validation was demonstrated for both the thermal stability of the Smart Capsule and the haemolysis detection of the minilysis device prototype taking the output of the Ultrospec2000 spectrophotometer as reference. Specifically, the onboard temperature test verified that the delivery system is capable of meeting the temperature-control requirements for the delivery. Even if the samples have been first manipulated to reach 11 °C before being inserted into the Smart Capsule, it proved to be effective in cooling down the samples and was able to maintain the temperature in the desired range during flight. On the other hand, in the laboratory haemolysis test, the Ultrospec 2000 spectrophotometer was shown to be competent as a quantitative reference for the development of the new minilysis device with a sequence of nine tests for seven different sub-bags and the dilution experiment during the last test. The trend of linear regression between the detection results from the spectrophotometer and minilysis device for all the original specimen and diluted ones during the last test confirmed the soundness of the minilysis device, whose output decreases with the increasing of haemolysis level.

## 5. Conclusions

The integration of an innovative haemolysis sensor device, and in future other medical sensors for pH, will be of utmost importance for the future of drone delivery, without wasting up to 30–45 min and the resources of medical facilities. This work represents a first step towards the implementation of such compact device on board a medical device for enabling drone delivery, such as the Smart Capsule. In the on-going design of the more compact version of the minilysis, the mechanical support will fit the shape of the cuvette to reduce the possible alignment error among the LED, specimen and photodiode. During future minilysis validation experiments, measurements will be performed with blood samples from several donors to show how the measurement results vary among different individuals. The final version of the minilysis is an optic device of haemolysis detection positioned directly on blood bags, which could be implemented to monitor the quality of blood and blood components during storage and delivery. As a future perspective, since lactate dehydrogenase(LDH) production is another well-known effect of haemolysis, one can figure out how to improve haemolysis detection by coupling the sensor to a fluidic or microfluidic chamber to perform a miniaturized enzymatic assay. LDH concentration is typically determined (e.g., for cell viability assays) by utilizing a coupled enzymatic reaction. Initially, LDH oxidizes lactate to pyruvate, then pyruvate reacts with a tetrazolium salt, namely iodonitrotetrazolium chloride, to form a chromogenic formazan compound with typical absorbance at 490 nm [30]. Consequently, the detection of an increased absorption at 490 nm can be correlated with an increased concentration of LDH, thus revealing haemolysis occurrence. Furthermore, another objective of future research is to explore the possibility of maintaining viable cultures of neurons or cardiac cells inside the Smart Capsule, which is already set up to maintain specific conditions for the tissues transported, recording their activity both in laboratory conditions and during autonomous flight. The types of signals evoked by cells in response to certain stimuli will be investigated, in order to classify them and use them in a controlled manner on board the Smart Capsules.

## Figures and Tables

**Figure 1 micromachines-13-01664-f001:**
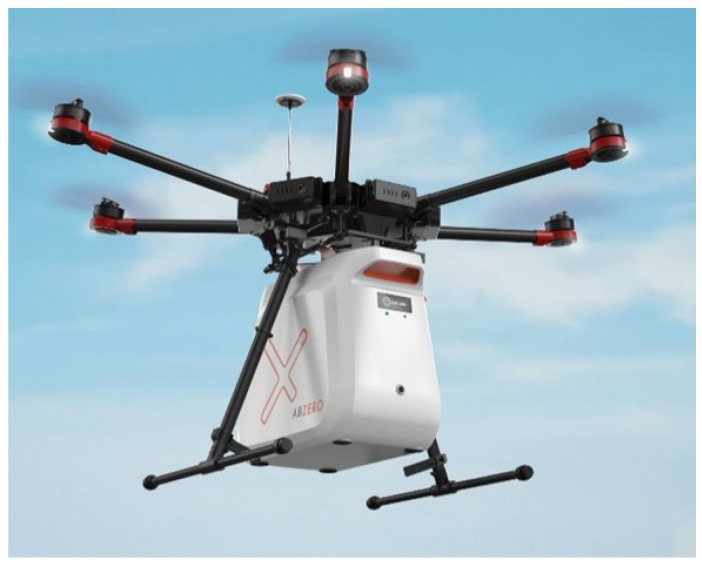
The Smart Capsule embedded on a drone.

**Figure 2 micromachines-13-01664-f002:**
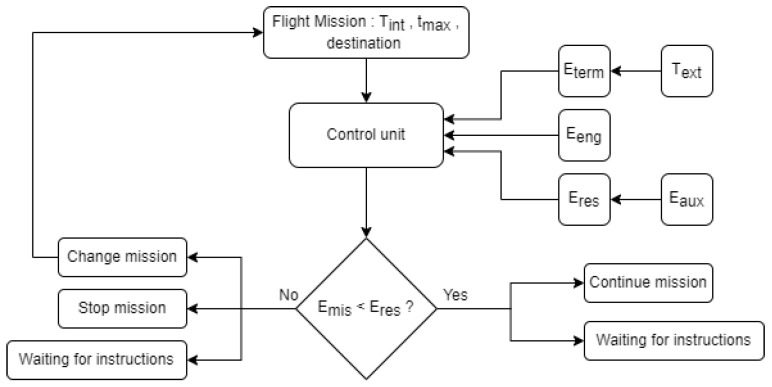
Schematic diagram of the temperature control unit.

**Figure 3 micromachines-13-01664-f003:**
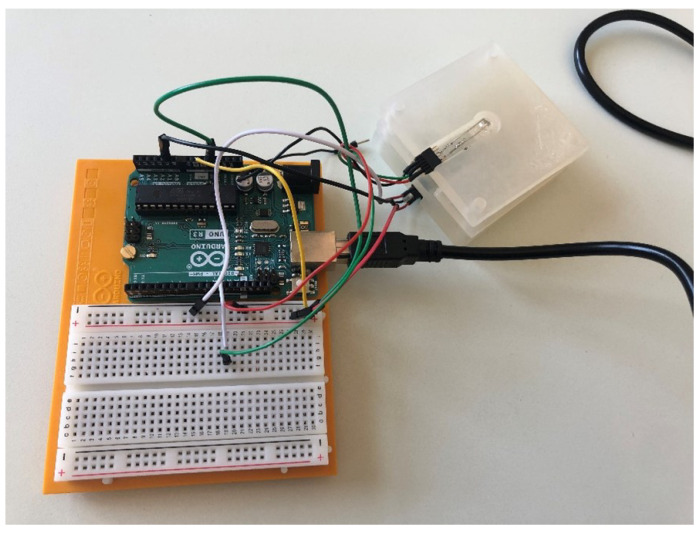
Minilysis concept validation prototype to measure the hemoglobin in the specimen, where the mechanical components were manufactured with 3D printer and the electronic components were connected on a microcontroller Arduino Uno, powered through a universal switching power supply converting 230 V AC to 7.5 V DC, which was then lowered to 5 V by the voltage regulator inside the Arduino Uno.

**Figure 4 micromachines-13-01664-f004:**
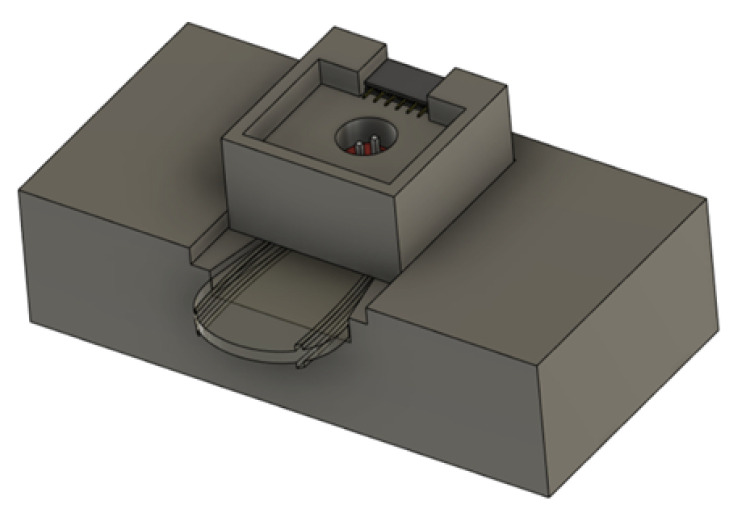
Mechanical design for a more compact minilysis on the platform of Fusion 360 Autodest.

**Figure 5 micromachines-13-01664-f005:**
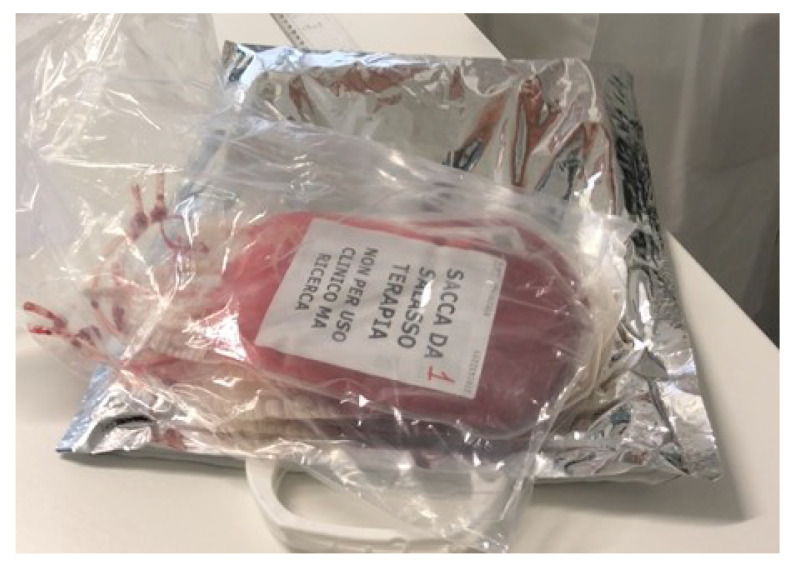
Blood donation bag from bloodletting therapy, which is labeled “bloodletting therapy bag, not for clinical use but for research” in Italian.

**Figure 6 micromachines-13-01664-f006:**
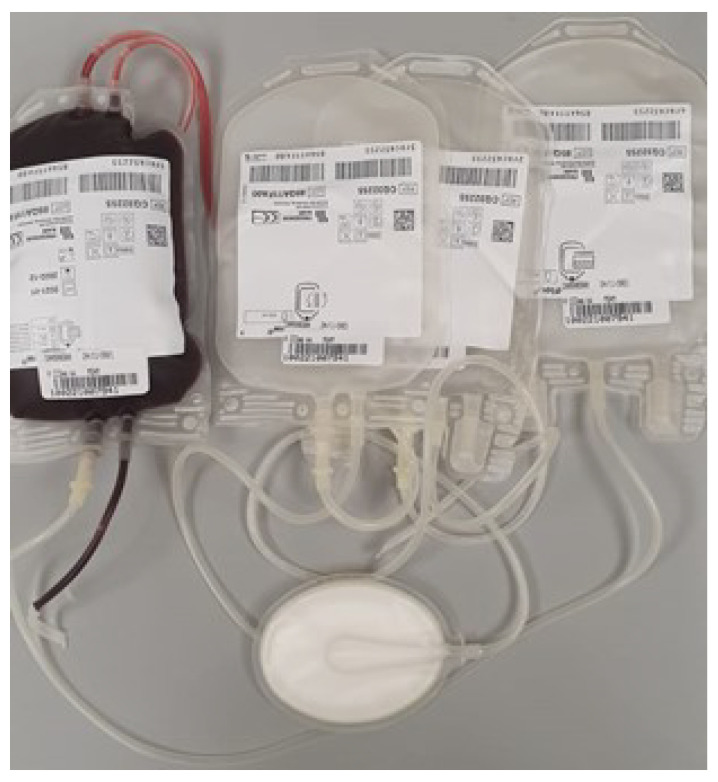
Blood separation to three bags, where the first bag was for plasma, the second one was for buffy coat, and the last one was to keep concentrated RBCs.

**Figure 7 micromachines-13-01664-f007:**
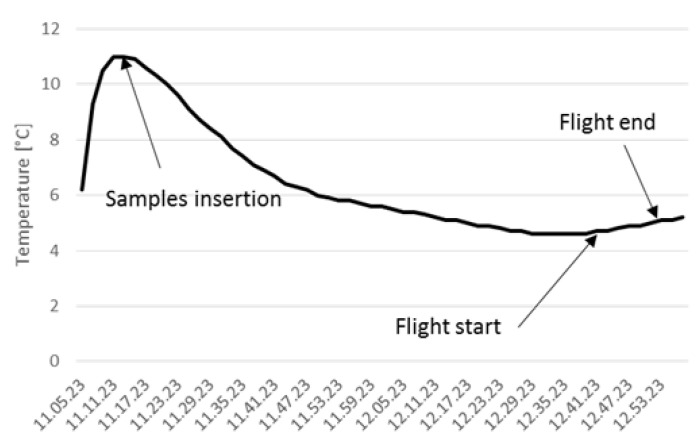
Temperature variation of the samples during the entire experiment, where the x axis represents the experimental time (h.min.s) with a fixed interval of 6 min.

**Figure 8 micromachines-13-01664-f008:**
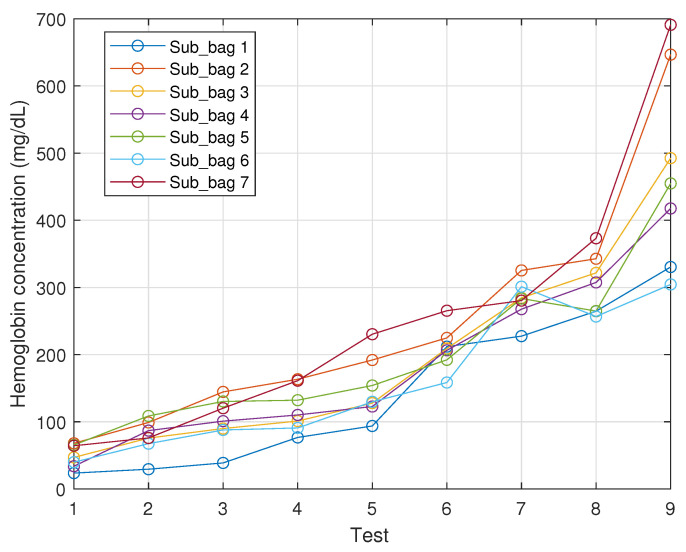
Hemoglobin concentration development of 7 sub-bags along 9 tests with the spectrophotometer.

**Figure 9 micromachines-13-01664-f009:**
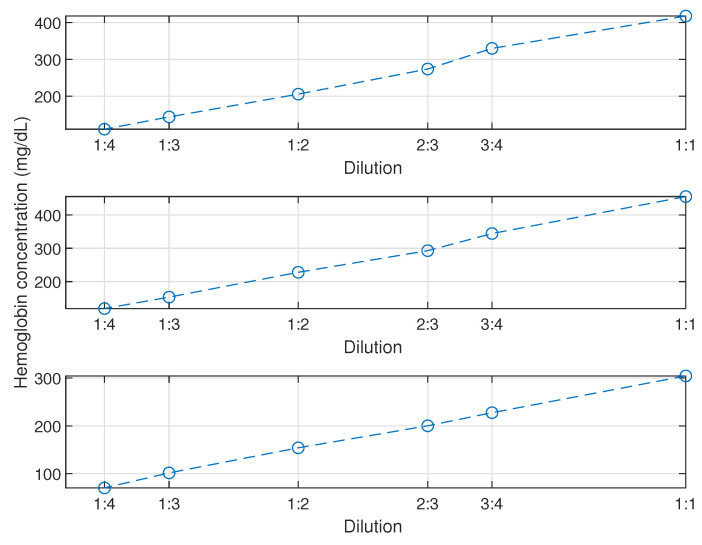
Hemoglobin concentration of dilution specimen for Sub-bags 4, 5 and 6 (from up to down) on the last day of test with the spectrophotometer.

**Figure 10 micromachines-13-01664-f010:**
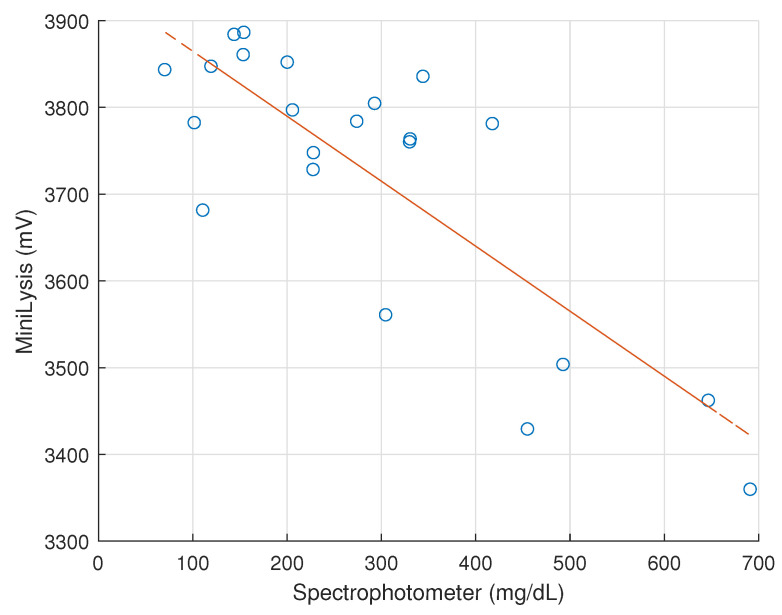
Linear regression relation between outputs of the minilysis prototype and the spectrophotometer.

## Data Availability

Not applicable.

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
