# Peer review of "Towards an Innovative Sensor in Smart Capsule for Aerial Drones for Blood and Blood Component Delivery"

_micromachines, 2022, doi:10.3390/mi13101664_

Round 1

Reviewer 1 Report (Previous Reviewer 3)

In the current version, the article can be published. All reviewer comments have been taken into account.

Author Response

Dear reviewer, we would like to express our sincere thanks to you for the positive comment on our updated manuscript.

Reviewer 2 Report (New Reviewer)

Dear Authors,

The research is impressive and the quality of work is satisfactory. I have listed my concerns in attached file in form of comments and highlights, pl refer. 

My major concerns:

1. Writing and references not upto the mark, recent studies are missed.

2. Novelty to be addressed better, especially the abstract is failed to depict.

3. Conclusion also needs revision.

4. Avoid long sentences and dragged lines. Add the relevant terms, few technical terms are not making any sense.

Author Response

Dear reviewer,

We would like to express our sincere thanks to you for the detailed and clear comments on our manuscript. We have revised the manuscript carefully according to your suggestions. Point-by-point responses are listed below.

Point 1: 1. Writing and references not up to the mark, recent studies are missed

Response 1: We are sorry in the version you have received, the reference is a mess. It has been updated.

Point 2: Novelty to be addressed better, especially the abstract is failed to depict.

Response 2: The abstracted has been modified to be more clear to demonstrate our novelty.

Point 3: Conclusion also needs revision.

Response 3: We have accepted your advice and divided the last section into "Discussion on results" and “conclusion” for a better understanding for readers.

Point 4: Avoid long sentences and dragged lines. Add the relevant terms, few technical terms are not making any sense.

Response 4: We have modified the long sentences you pointed and unified the terms you highlighted.

Besides, the title has been modified as “Towards an innovative sensor in Smart Capsule for aerial drones for blood and blood component delivery”; Some typos have been also corrected. Some brief explanation about the structure and dimension have been added for the more compact version as requested.

This manuscript is a resubmission of an earlier submission. The following is a list of the peer review reports and author responses from that submission.

Round 1

Reviewer 1 Report

Good morning authors,

The paper submitted is interesting and has several novel aspects. It is publishable but after revisions. One of the central questions that needs to be answered in the revised manuscript is that if blood analysis technology can be miniaturised and conducted relatively cheaply, why is a smart capsule required, why can't the blood be analysed at the point of care, where the drone lands?

Here are some specific points that need to be addressed before the paper can be published.

Line 6 please edit this sentence as the number two is used and it makes the prose a little bit confusing.

A general comment concerning the abstract. It's a little bit long so it would benefit from some editing but also the inclusion of some metrics or example or summary quantitative data. The novel part of the work centres around the innovative haemolysis and pH sensors that are on board therefore this parts of the work should have the priority in the abstract. The abstract would also benefit from a clear unambiguous aim or hypothesis that will be tested.

The beginning of the introduction, please confirm that blood and blood components are classified as essential medicines comment on these medical products?

The paper is written well but there are one or two grammatical errors, see line 30. This could be corrected to ....The average adult has about 5 L of blood circulating........ thus please proof read the rest of the manuscript carefully to pick up similar slight grammatical improvements just before publication.

Line 70 please edit the bracketed question mark.

Line 83 please define what is meant by temperature controls unit. This implies that both temperature can be monitored but also adjusted, i.e. the smart capsule is an active thermostatically controlled cooling unit. However on further reading of the paper the smart capsule is an insulated transport box with accurate temperature monitoring. Thus in the revised manuscript please define the terms of temperature control that are used in the paper.

The introduction should finish with a clear focus to aim or a hypothesis that will be tested.

Section 2.1. The paper would benefit from more technical details of the smart capsule. Perhaps even a technical drawing of its construction. This should indicate the dimensions, the type and depth of the polyurethane insulation and exactly where the temperature monitoring devices are placed.

Has this box been crash tested? 

If the details in innovation concerning the smart box have already been published into papers then this is not novel, it's excellent technology that the paper should focus on the on board blood monitoring rather than the previously published details concerning the smart box. Please edit the revised manuscript accordingly.

Lines 126 to 127. I was solutions of water and paraffin mixtures being used this phase change materials, i.e. cooling blocks? If so please describe them as such.

More information is required concerning the parameters described in the excellent schematic given in figure 2. For example please define what the auxiliary energy source is? Is this the energy required to change the phase in the phase change material used as a cooling block.

The design and production of the more compact version of the minilysis equipment is the novel part of the work and therefore the paper should really be dedicated to providing insights to its development and also evaluation of its performance. Therefore please describe in the revised manuscript how the modified analysis equipment was benchmarked to accepted regulatory standards. 

Please provide some statistical analysis of the on-board temperature tests, was the mean kinetic temperature measured for example? Also to increase the number of repeats why couldn't the smart box be placed in a series of different temperature environments without the need for conducting drone flights? This could have provided a lot of thermal data that could have given a better statistical foundation and confidence in the excellent technology that has been developed.

The laboratory tests for the haemoglobin did have a good number of repeats and statistical significance.

Lines 261 and 262. From the description it appears that the minilysis device was used only on the last test day for all the original specimens and the diluted ones. Therefore could time have affected the correlation between the standard laboratory test and the minilysis device?

The correlation given in figure 10 is encouraging but it has quite a lot of scatter. What a box plot be useful? Or the addition of the two standard deviation confidence levels plotted above and below the correlation line? How does this correlation map onto expected confidence from the medical regulator?

Concerning the issue of maintaining the position of the cuvette describe the lines 267 to 269 could a 3D printed addition to the holder be of use? And why wasn't a pilot study introduced before the analysis to check for cuvette positioning? Please evaluation this issue in the discussion.

Line 274, apologies if I've missed the evidence for this, but where is the data that supports the implementation of AI-based technology? An algorithm or data needs to be given in the manuscript to provide supporting evidence?

Line 293 describes the compact nature of the device. Thus in the revised paper please provide the dimensions, the volume and the mass of the new minilysis device. 

Also a general point, how is the minilysis device powered? And is there are enough battery capacity in a typical drone to allow it to be used?

Please add a conclusion to the revised manuscript.

Reviewer 2 Report

This is the third revision of the manuscript with now new title and new submission (old submission 1809208). Unfortunately, no significant progress is evident on the main point of criticism, the scientific content. Please look at my comments on the previous manuscript (1809208). Unfortunately, I have to remain with my last statement: “The scientific content and the novelty compared to the state of the art remain questionable, unclear and insufficient”. With only textual changes in the manuscript it is not done. More research work (analysis, measurements, calculations, ...) is needed to justify a publication in a scientific journal. Often the manuscript talks about the future, like integrating a pH-meter or the minilysis into the drone. Why is this not done and its effectiveness verified in a study? Even the title includes pH sensor although no work has been done in this direction. I cannot recommend the current status of the work for publication.

Reviewer 3 Report

1. It should be more emphasized that the need for blood evaluation and tests lasting more than 30 minutes often makes up for the difference in transportation time and raises questions about the viability of drone transport.

2. It is also necessary to analyze legal issues related to blood transport in selected countries . In my opinion, it is necessary to critically address this issue after checking the legal considerations for blood transport in specific countries.

4. Specifically, haemolysis inside the body can be caused by a large number 69 of medical conditions [? ], - missing citation number

5. Statistically, the correlation 260 coefficient is -0.8085, with a p-values of 0.0000

Please correct p<0.0001

6. Conclusions are missing in main text